# miRNA expression in advanced Algerian breast cancer tissues

**Mohamad Ali Tfaily[1‡], Farah Nassar[1‡], Leila-Sarah Sellam[2], Zine-Charaf Amir-Tidadini[3], Fatima Asselah[3], Mehdi Bourouba[2]\*, Nasr Rihab[1]\***

**1** Department of Anatomy, Cell Biology and Physiological Sciences, Faculty of Medicine, American University of Beirut, Beirut, Lebanon, **2** Department of Cell and Molecular Biology, Team Cytokines and Nitric oxide synthases, Faculty of Biology, University of Sciences and Technology Houari Boumediene USTHB, Algiers, Algeria, **3** Central Laboratory for Anatomopathology, Mustapha Pacha Hospital, Algiers, Algeria

‡ These authors share first authorship on this work.
\* rn03@aub.edu.lb (RN); mbourouba@usthb.dz (MB)

## Abstract

Breast cancer is one of the commonest cancers among Algerian females. Compared to Western populations, the median age of diagnosis of breast cancer is much lower in Algeria. The objective of this study is to explore the expression of several miRNAs reported to be deregulated in breast cancer. The miRNAs miR-21, miR-125b, miR-100, miR-425-5p, miR-200c, miR-183 and miR-182 were studied on tumor and normal adjacent Algerian breast tissues using quantitative reverse transcription real time PCR, and the results were analyzed according to clinical characteristics. Compared to the normal adjacent tissues, miR-21, miR-183, miR-182, miR-425-5p and miR-200c were found to be upregulated while miR-100 and miR-125b were insignificantly deregulated. A positive correlation was noted among miR-183, miR-182 and miR-200c and among miR-425-5p, miR-183, miR-200c and miR-21. Further global miRNA microarray profiling studies can aid in finding ethnic specific miRNA biomarkers in the Algerian breast cancer population.

## 1. Introduction

Breast cancer is the most common cancer affecting females worldwide, with more than 2 million new cases diagnosed in 2018. In the Middle East and North Africa (MENA) region, breast cancer constitutes 31.1% of the total cancer incidence in females, with a mortality rate of 20.9%. In Algeria, the number of cases diagnosed in 2018 reached 11847, constituting 24% of the cases of cancer incidence among Algerian females in 2018, which is a rate much higher than the rest of the MENA region [1,2]. The median age of diagnosis was found to be 48, and 66% of the diagnosed Algerian females were below the age of 50. This age is more than a decade earlier than that of Western Europe and the United States of America [3–5].

There is no available information concerning the severity of the disease in Algeria [3, 4] except that it constituted 13% of the total cancer mortality among the Algerian population in 2018 [2]. Reports show that breast cancer rates also vary between regions. The crude incidence

**Competing interests:** The authors have declared that no competing interests exist.

of breast cancer was 20.5 in Sétif between 2003–2007 while it reached 35.2 in Annaba between 2007–2009 [6]. It is noteworthy to mention that the diagnosis of all types of cancers in Algeria is usually late in over two-thirds of the cases. Many cases of death due to breast cancer have been reported primarily due to the ineffective screening methods that lead to delayed diagnoses. Efforts have been put into increasing the screening rates by introducing a mobile mammography that can help cover the vast portion of Algerian land especially the rural areas that happen to be sparsely distributed in this largest country in Africa [7].

The young median age at diagnosis, high incidence and mortality rates and the inaccessibility of current screening tools of breast cancer highlight the importance of novel screening techniques and early detection in decreasing the morbidity and mortality of the disease. Novel biomarkers including circulating miRNA levels are under extensive study for their potential role in being effective screening tools of this disease.

microRNAs (miRNAs) are a subclass of noncoding RNA molecules that were discovered in *C. elegans*. This subclass leads to gene modulation at the post-transcriptional level [8]. They are transcribed from a miRNA in several steps to give rise to mature miRNA incorporated into the RNA-induced silencing complex [9]. Disrupted miRNA homeostasis has been delineated in several diseases including cardiomyopathies, cancers, diabetes and neurodegenerative disorders [10, 11]. The first evidence of miRNA involvement in human cancer came from its deregulation in Chronic Lymphocytic Leukemia [12]. This finding was proceeded by proof of deregulation of miR-143 and miR-145 in colon carcinomas and miR-125b, miR-145, miR-21 and miR-155 in breast cancer tissues [13, 14]. Moreover, more than 50% of miRNA genes lie on chromosomal regions that are altered in cancer pathogenesis which explains the involvement of miRNAs in human cancers [15]. Numerous studies have correlated miRNA deregulation to cellular processes involved in modulation of tumor suppressor genes and oncogenes through cell cycle regulation and apoptosis [16].

The strong correlation between miRNA dysregulation and different cancers made their role revered as biomarkers especially that miRNAs can also be found in the plasma and other biological fluids such as the urine and cerebrospinal fluid [17, 18]. This suggests that circulating miRNAs are practical detectable tumor biomarkers especially that their dysregulation is reflective of that of the tumor tissue [19].

The deregulation of miRNAs such as miR-21, miR-425-5p, miR-183, miR-182, miR-200c, miR-125b and miR-100 has been studied extensively in the literature, as well as in breast cancer tissues of neighboring Arab populations as in Lebanon [20]. The latter study is the only array available for breast cancer tissues in the MENA for the Arab population and it shows that miR-183, miR-miR-182, miR-200c, miR-425-5p and miR-21 were significantly upregulated while miR-125b and 100 were significantly downregulated in tumor versus normal adjacent tissues. In this study, we aim to explore the deregulation of these miRNAs in breast cancer formalin-fixed paraffin-embedded (FFPE) tissues of Algerian patients.

## 2. Materials and methods

### 2.1. Tissue specimen

The Institutional Review Board of the Mustapha Pacha Hospital approved the study. All research was performed in accordance with relevant guidelines and regulations. Analyzed breast cancer tissues were obtained from leftover tissues, remnants of specimens collected for diagnosis from patients being treated between 2011 and 2012 at M. Pacha Hospital. The studied samples were received coded with no identifiers and were prepared at the central anatomo-pathology laboratory for routine diagnostics and stored at the hospital tissue bank. Clinical

and pathological data including age at diagnosis, ER status, PR status, and HER2 over-expression were available for all samples included in this study.

## 2.2. Total RNA extraction

Total RNA from 22 tumor and 8 normal adjacent FFPE tissues was extracted using the protocol of RecoverAll Total Nucleic Acid Isolation Kit for FFPE samples (Ambion, USA). Deparaffinization of the FFPE samples was first done using xylene at 50˚C. The xylene was then removed by washing the samples twice with 100% ethanol. In order to digest the proteins, the samples were incubated with protease enzyme at 50˚C for 15 minutes and then at 80˚C. Total RNA was then captured through glass-fiber filter columns proceeded by washing with high ethanol-wash buffers. DNA was digested by DNase for 30 minutes which was followed by washing and eluting steps to isolate RNA. The quality and concentration of the purified RNA were evaluated by means of the Nanodrop ND1000. Then RNA was stored at -80˚C.

**miRNA expression by quantitative Real Time Polymerase chain reaction (RT-qPCR).** Using the instructions of TaqMan MicroRNA Reverse Transcription Kit (Applied Biosystems, USA), 10 ng of the total RNA were reverse transcribed. Small nucleus RNA RNU6B, hsa-miR-125b, hsa-miR-425-5p, hsa-miR-21, hsa-miR-200c, hsa-miR-183, hsa-miR-182 and hsa-miR-100 probes and primers were ordered as part of the TaqMan microRNA Assays Kit (Applied Biosystems, USA). cDNA synthesis was performed in a multiplex reaction in which two miRNA primers were used along with the endogenous control (RNU6B). RT-qPCR was performed by means of BioRad CFX384 Real Time System, C1000 Thermal Cycler (Germany). Each well consisted of 5 µL of 2x TaqMan Universal Master Mix with no Amperase Uracil N-glycosylase (UNG) (Applied Biosystems, USA), 2 uL of RNase free water, 0.5 µL of corresponding 20x miRNA probe and 2.5 µL of the cDNA. The reactions were completed in duplicates for each microRNA probe.

Tumor tissues were normalized according to the normal adjacent tissues in the same RT-qPCR run to ensure inter-run calibration. The conditions of cycling were 95˚C for 10 minutes and 40 cycles of 95˚C for 15 seconds and an annealing temperature of 60˚C for 60 seconds.

By means of the ΔΔCt equation, the expression of experimental miRNA in tumor tissues was calculated in comparison to the normal adjacent tissue (NAT) samples using the endogenous control RNU6B.

## 2.3. Statistical analysis

After checking for normality using Kolmogorov Smirnov test, the samples were not found to follow a normal distribution. Spearman's correlation was used to correlate the different variables. The Wilcoxon signed rank test was used to compare the means between miRNA expression in tumor tissues and NAT. Mann Whitney test was used to compare fold change in expression and HER2 status. Statistical analysis was performed using GraphPad Prism and SPSS software package version 25. Heatmap analysis were performed using Genesis software.

## 2.4. Bioinformatics analysis

Network Analysis between the miRNAs was performed using Pathway Studio which enables the analysis and visualization of altered pathways required to construct and recognize altered cellular processes and involved molecular functional pathways in breast cancer. Enrichr software (https://amp.pharm.mssm.edu/Enrichr/enrich) was used to identify the GO Biological Processes of miRNA validated targets.

# 3. Results

## 3.1. Baseline demographics

The clinical and pathological data of 20 FFPE tumor samples taken from Algerian breast cancer patients have been obtained. 75% of the tumor samples are Estrogen Receptor/progesterone receptor positive (ER/PR+) and 55% have a Human Epidermal Growth Factor Receptor 2 positive (HER2+) status. 45% of the samples were from patients above the age of 50. All analyzed patients had advanced breast cancer tumors (Stages III or IV) at the time of diagnosis.

## 3.2. miRNA expression in breast cancer tissues and normal adjacent tissues

miR-21 (p<0.0001), miR-183 (p<0.0001), miR-182 (p<0.0001), miR-200c (p<0.0001) and miR-425-5p (p = 0.003) were found to be significantly overexpressed in tumor tissues while miR-100 and miR-125b were insignificantly deregulated compared to the NAT (Figs 1 and 2). Heatmap analysis revealed different clustering between cancer and NAT samples. Within the cancer tissues, two subclusters that differed in HER2 status were obtained (Fig 3).

The variation of miRNA expression according to HER2 status (HER2+ vs HER2-) was examined using Mann-Whitney test. miR-21 and miR-125b were found to be significantly upregulated in HER2- samples (p-value = 0.0031 and 0.031 respectively) as compared to HER2 + samples (Fig 4A and 4B). No significant change in expression with HER2 status was noted with the other miRNAs. Similarly, expression of miRNA was compared between patients below and above 50 years of age. No significant dysregulation of the miRNAs was found. No analysis was performed on the ER grouping as the number of ER negative samples is not significant.

## 3.3. Correlation between different miRNAs

Spearman analysis was performed to examine the relationship between different miRNA expressions. miR-183 was found to have a significant positive correlation with miR-182, miR-425-5p, miR-200c and miR-21 (correlation coefficient = 0.897, 0.694, 0.798, 0.74 respectively, p-value <0.001). This indicates that these miRNAs can be expected to be upregulated along with miR-183.

In addition, expressions of miR-182 and miR-200c were found to be positively correlated with each other with a correlation coefficient of 0.792. miR-425-5p expression was correlated with that of miR-200c and miR-21 with a correlation coefficient of 0.756 and 0.850 respectively and p-value less than 0.01. miR-21 and miR-200c are moderately correlated with a correlation coefficient of 0.655.

## 3.4 miRNA deregulation in our study compared to the literature

miRNA deregulation in the Algerian population was compared to that found from several other studies in the literature (Table 1). Our results revealed upregulation of miR-21, miR-183, miR-182, miR-425-5p and miR-200c that is in concordance with the results across the literature. Nevertheless, our study revealed non-significant upregulation of miR-125b and miR-100 while results in the literature reveal downregulation of these two miRNAs.

# 4. Discussion

In this study, we explored the expression of 7 miRNAs that were previously reported in the literature to be significantly dysregulated in breast cancer tissues. This is the first study to be done on North African tissue samples, and Algerian samples specifically. The studied miRNA (miR-125b, 183, 182, 21, 125b, 200c and 425-5p) are involved in several vital biological

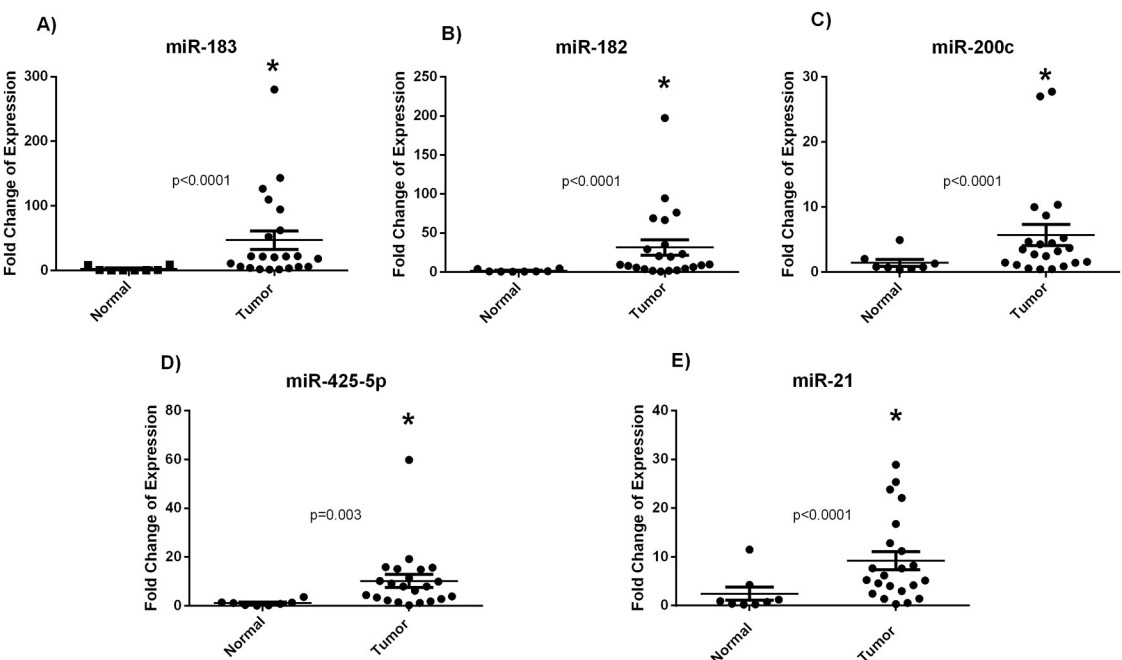

**Fig 1. Upregulated miRNA expression in Algerian breast cancer tissues.** Dot plots show the fold change of expression of miR-183, miR-182, miR-200c, miR-425-5p and miR-21 in tumor breast cancer tissues compared to the normal adjacent tissues, obtained using RT-qPCR with RNU6B as an endogenous control. * signifies p<0.05, using Wilcoxon signed rank test.

processes of breast cancer which are cell differentiation, cell motility, cell death, oncogenesis, cell invasion and migration, DNA damage and chemosensitivity (Fig 5 and S1 Fig). miR-21, miR-183, miR-182, miR-200c and miR-425-5p were found to be significantly upregulated in breast cancer tissues relative to the NAT. Each of these miRNAs was found to be correlated in terms of its upregulation with the other miRNAs. Only miR-21 and miR-125b expressions

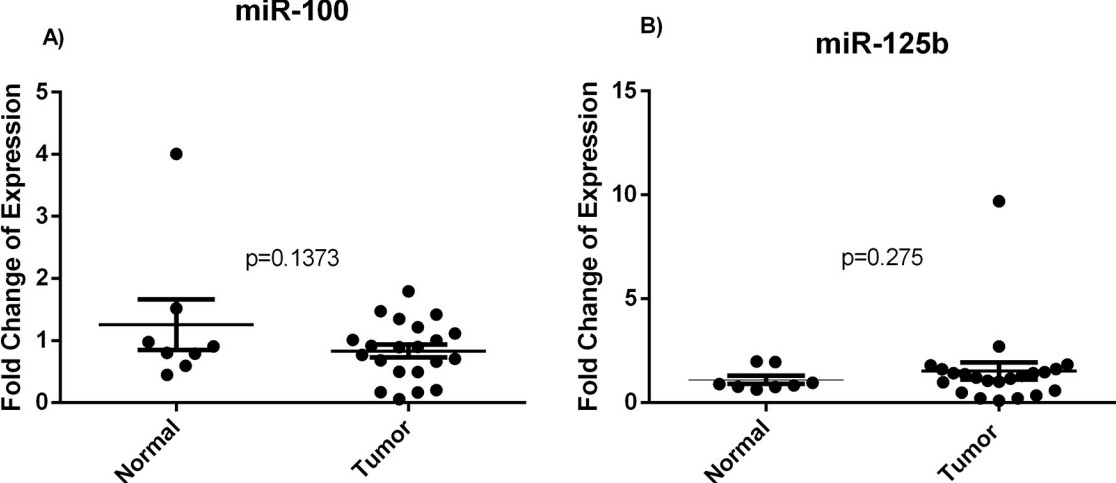

**Fig 2. Deregulated miRNA expression in Algerian breast cancer tissues.** Dot plots show the fold change of expression of miR-100 and miR-125b in tumor breast cancer tissues compared to the normal adjacent tissues, obtained using RT-qPCR with RNU6B as an endogenous control.

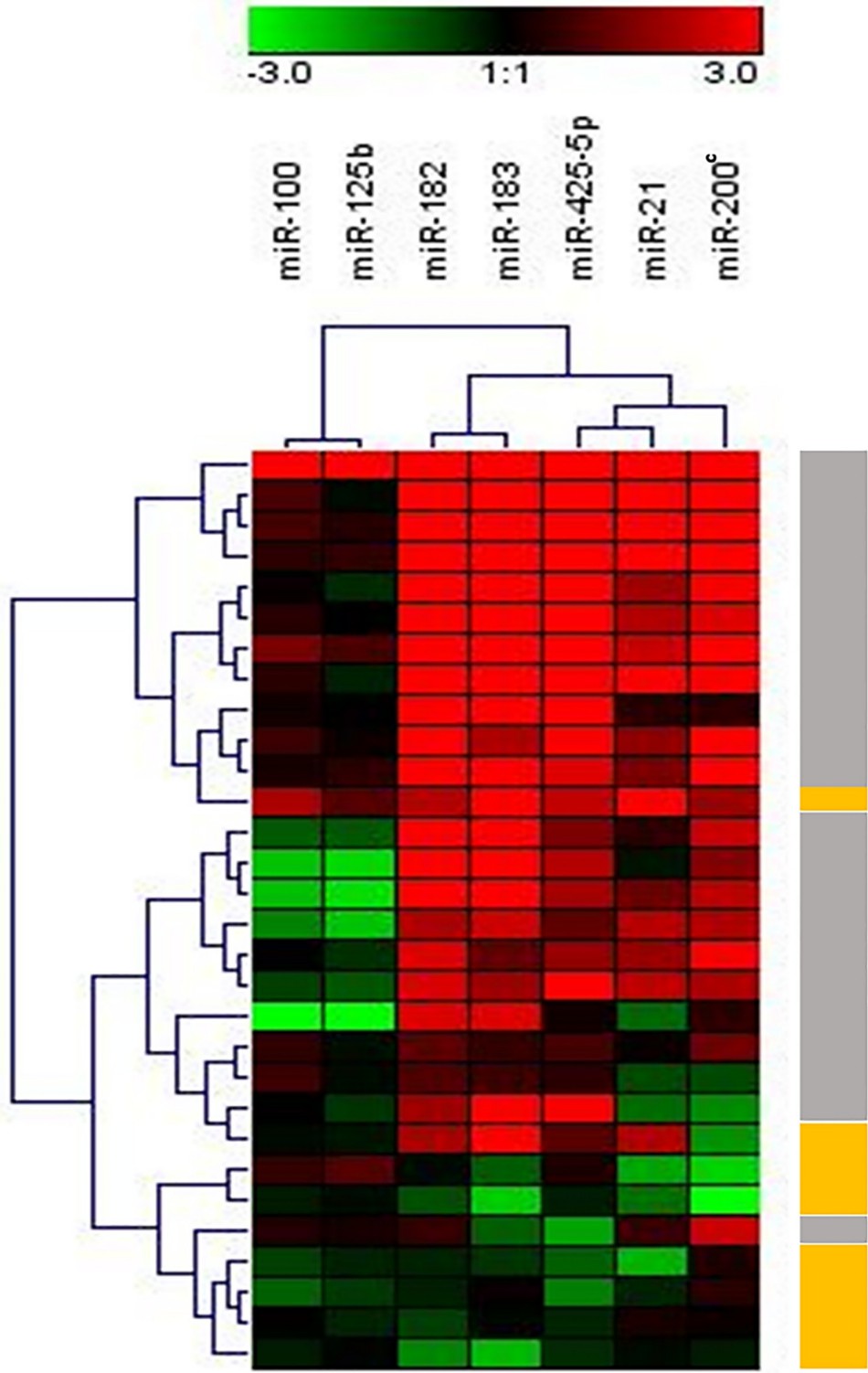

**Fig 3. Heatmap and dendogram analysis of miRNA expression in normal and BC tissues.** Fold expression data were Log2 transformed before analysis. miRNA profiling segregated the tumor tissues and the normal adjacent tissues. Left-hand side dendrogram corresponds to a hierarchical clustering of the tissue samples. The upper side dendrogram corresponds to a hierarchical clustering of miRNA expression in the FFPE tissues. Grey boxes denote tumor samples and yellow boxes denote normal adjacent tissues.

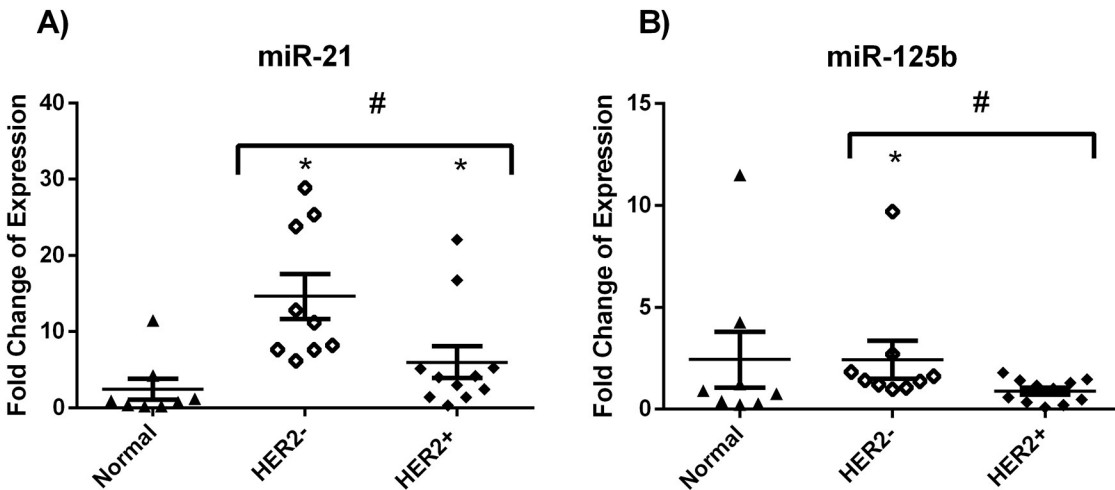

**Fig 4. Significant deregulation of miR-21 (A) and miR-125b (B) expression with HER2 status.** Dot plots represent the fold change of expression of miR-21 in HER2+, HER2- and normal adjacent tissues. *denotes p<0.05 for tumor versus normal using Wilcoxon signed rank test. #denotes p<0.05 using Mann-Whitney test.

were found to be significantly upregulated with a HER2 negative status. miR-100 and miR-125b were found to be insignificantly deregulated relative to the normal adjacent tissue. Moreover, these two miRNAs shared a correlated pattern of expression.

miR-21 was upregulated in the breast cancer tissues relative to the NAT, and it was found to be positively correlated with a HER2- status. Our data of miR-21 was consistent with the literature where it is usually found to be significantly upregulated in breast cancer [13]. This is explained by the fact that this miRNA has been shown to have an oncogenic role. For instance, the function of miR-21 has been associated with tumor suppressor gene p53; miR-21's downregulation in a large number of cancers leads to a decrease in cell growth. Thus it is expected to be upregulated in the tissues under study [21]. Upregulation of miR-21 was found to be correlated with a HER2- status in Algerian samples. This correlation was insignificant in a study by Nassar *et al.*[22]. Higher miR-21 has been reported in HER2+ compared to HER2- in FFPE tissue samples collected from 15 patients who had undergone surgery for primary breast cancer [23]. In an in vitro study, a MEK-ERK pathway induced miR-21 expression downstream of HER2/neu gene in breast cancer cells. In HER2/neu negative breast cancer cells, overexpression of MEK1/2-ERK1/2 pathway activators, H-Ras (G12V) and ID-1, also significantly increased the levels of miR-21 so miR-21 expression could be affected by the MEK1/2-ERK1/2 pathway activators independent of HER2 expression [24].

**Table 1. Mode of deregulation of the studied miRNA in different studies in comparison to our findings.** (Up means upregulated; Down means downregulation).

| miRNA | Deregulation in Our study | Deregulation in Literature | Population/Country if specified | References |
|---|---|---|---|---|
| miR-183 | Up | Up | Lebanese, American, Taiwan | [20, 25, 31] |
| miR-182 | Up | Up | Lebanese, Chinese | [20, 32, 55] |
| miR-125b | Up | Down | Lebanese, Spanish, American | [20, 38] |
| miR-100 | Up | Down | Spanish, Austria | [38, 46, 47] |
| miR-200c | Up | Up | Lebanese, American, Taiwan | [20,25,26] |
| miR-425-5p | Up | Up | Lebanese, American | [20, 25] |
| miR-21 | Up | Up | Italy, USA, Taiwan | [13, 25, 54] |

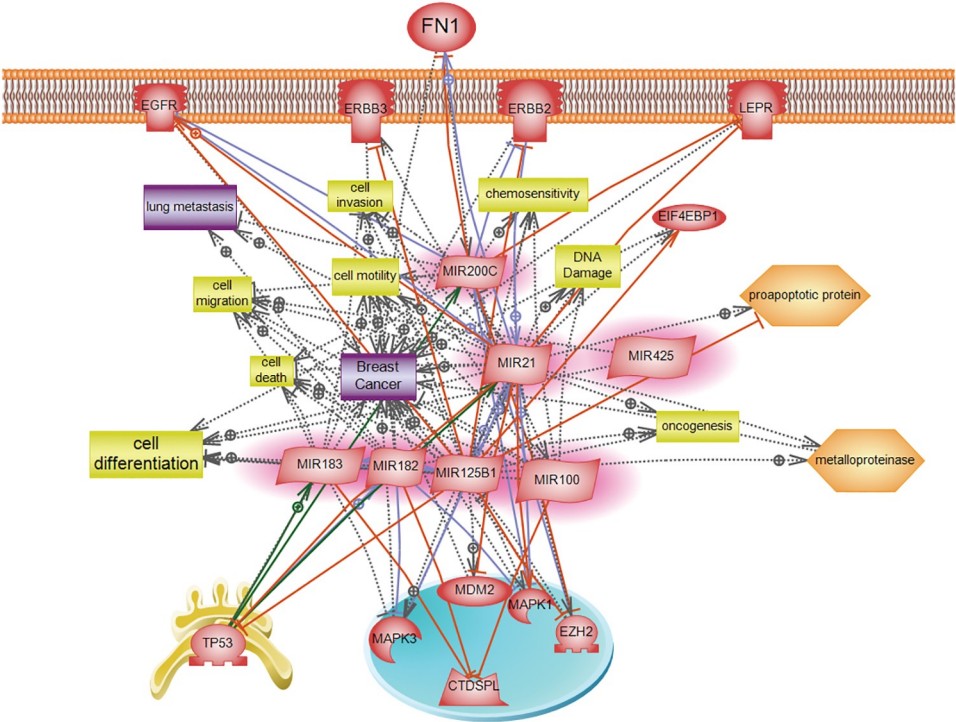

**Fig 5. Pathway studio network analysis of the studied microRNA in breast cancer.** A biological network was created using the Pathway Studio 9.0 program to visualize the role of different microRNA discussed in this manuscript in chosen breast cancer-related pathways.

miR-200c upregulation in our study coincides with the results by Nassar *et al.* in Lebanese and American patients and several other studies in the literature [25, 26]. Its upregulation, nevertheless, was associated with increased sensitivity to radiation treatment, and it is associated with better overall survival in ER positive breast cancer [27, 28]. miR-200c belongs to a cluster of miRNAs known as the miR-200bc/429 cluster. By reducing the expression of p27/kip1 and by upregulating inhibitory phosphorylation of Cdc25C gene, this cluster causes G2/M cell cycle arrest [29]. Moreover, miR-200c functions by suppressing epithelial-mesenchymal transition, reducing cell viability, increasing apoptosis and inhibiting the migration and invasion of breast cancer cells [30].

An upregulation of both miR-183 and miR-182 was found in the Algerian breast cancer tissues. miR-183 upregulation coincides with the results of a study by Chen et al. where miR-183, in addition to miR-21 and miR-200c were find to be significantly upregulated in the FFPE breast cancer tissues[25]. This was also noted in a study on Lebanese FFPE breast cancer tissues [20, 31]. miR-182 upregulation was noted in more than one study where its fold change of expression is usually more than 4 folds higher in cancerous breast tissue than in paracancerous tissue [32]. We found that the upregulation of miR-183 and miR-182 were positively correlated. This can be explained by the fact that these miRNAs are located in the same gene cluster [33]. The cluster of miR-183/182/96 has been found to have an upregulated expression in breast cancer tissues. Their overexpression was correlated with tumor, node and metastasis (TNM) stage and possible metastasis [33]. In our study, no correlation was found between miR-183 or miR-182 with HER2 status and this is in accordance with a study by Li *et al.*[34]. miR-183/-96/-182 cluster promoted rapid completion of mitosis thus increasing cell proliferation [34]. miR-183, as an oncogene in breast cancer, represses the expression of EGR1 [35].

Moreover, it was shown to inhibit cell migration by repressing Ezrin gene in breast cancer cell line T47D [36]. miR-182 promotes invasiveness by regulating formation of filopodia and distribution of actin in breast cancer cells [37].

In our study, both miR-125b and miR-100 showed no significant difference between cancerous tissue and normal adjacent tissues. This does not concur with other studies in the literature that found miR-125b expression to be significantly downregulated in cancerous tissue relative to the normal adjacent tissue [38]. miR-125b has a tumor suppressor role by targeting the 3' untranslated region (UTR) of mRNA of glutamyl aminopeptidase encoding gene ENPEP, as well as mRNA of Casein Kinase 2-α (CK2-α) [39] which are involved in breast cancer tumorigenesis [40–42]. Furthermore, increased expression of miR-125b in mammary cells caused a decrease in cell proliferation by inducing cycle arrest at the G2/M phase and reducing anchorage-dependent cell growth of mammary cells [39]. In addition, miR-125b targets ARID3B gene which when silenced, decreases cellular proliferation [43]. This provides further proof of the tumor suppressor role of miR-125b in breast cancer. In our study, miR-125b was found be to be upregulated with HER2- status which does not concur with the results in the literature where miR-125b was found to be significantly upregulated with HER2+ status [44].

Concerning miR-100, the literature shows varying results concerning its deregulation. In a study analyzing TCGA data, miR-100 was found to be downregulated in all breast cancer subtypes [45]. miR-100 was found to be downregulated in breast cancer cells, which led to an increase in the insulin-like growth factor 2 (IGF2) expression [46]. miR-100 induction was found to have a slight stimulatory outcome on growth of the SK-BR-3 cells, but had a serious damage on breast cancer cells. Silencing miR-100 had an apoptotic effect on breast cancer cell line SK-BR-3 thus causing tumor suppression both in vivo and in vitro [47]. The present study showed an insignificant miR-100 deregulation, which may also be explained by the small sample size. miR-125b and miR-100 in the Algerian breast cancer tissues were found to be correlated as miR-125b and let-7a are distant miRNAs that reside on the same fragment, as per expressed sequence tag evidence [48].

A study by Nassar et al. done on breast cancer tissues of Lebanese patients showed similar results as the present study in terms of upregulation of miR-21, miR-425-5p, miR-200c and miR-183 and miR-182. The downregulation of miR-125b and miR-100 in the present study, to the contrary of study on samples of Lebanese patients, did not reach significance. This similarity is expected as both Algerian and Lebanese patients have a similar ethnic profile and lie within the Middle East and North Africa region [20]. A limitation of the study is the small sample size that did not offer us adequate power to study relationships between the clinical characteristics and miRNA deregulation. In addition, it was difficult to obtain some of the clinical characteristics of some patients due to cross country and institutional barriers.

miR-21, miR-183, miR-182, miR-425-5p and miR-200c were found to be upregulated in tumor versus normal adjacent breast tissues, similar to the Lebanese patients and patients in other studies reported in the literature [20, 25]. This study is the first to report the patterns of miRNA dysregulation in the Algerian population, one of North African populations that are known to possess great gene pool heterogeneity from Eurasia, Sub Saharan Africa and North Africa [49]. Furthermore, several meta-analyses report ethnic differences in miRNA deregulations in cancer [50, 51]. Most of the studies on miRNAs as biomarkers were performed on Asian and European populations, with very few being performed on populations of African descent. Data from these studies showed different expression of miRNA between African-Americans and European-Americans in breast cancer and lung cancer, respectively [52, 53]. The aim of this pilot study was to lay ground to microRNA research in the region. As such, further global miRNA microarray profiling can aid in finding ethnic specific miRNA biomarkers in the Algerian breast cancer population. Moreover, further research on circulating microRNA

is needed which along with the availability of advanced infrastructure, technology and training might serve identifying potential biomarkers for early detection of breast cancer.

## Supporting information

**S1 Fig. GO Biological Processes of validated targets for miR-182 (A),183 (B), 21 (C), 200c (D), 125b (E), and 100 (F).** Validated targets were identified using PubMed literature search and their GO Biological Processes were determined using Enrich software ([https://amp.pharm.mssm.edu/Enrich/enrich](https://amp.pharm.mssm.edu/Enrich/enrich)).
(TIF)

## Acknowledgments

The authors acknowledge Dr. Firas Kobeissy for his help with Pathway Studios software, and the technical help of the Basic Research Core Facilities at the Faculty of Medicine at AUB. Dr. Rihab Nasr is a member of International Breast Cancer and Nutrition (IBCN).

## Author Contributions

**Conceptualization:** Mehdi Bourouba, Nasr Rihab.

**Data curation:** Farah Nassar.

**Formal analysis:** Mohamad Ali Tfaily, Farah Nassar, Mehdi Bourouba, Nasr Rihab.

**Methodology:** Mohamad Ali Tfaily, Farah Nassar, Zine-Charaf Amir-Tidadini, Fatima Asselah, Mehdi Bourouba, Nasr Rihab.

**Project administration:** Nasr Rihab.

**Resources:** Leila-Sarah Sellam, Zine-Charaf Amir-Tidadini, Fatima Asselah, Mehdi Bourouba, Nasr Rihab.

**Supervision:** Farah Nassar, Nasr Rihab.

**Writing – original draft:** Mohamad Ali Tfaily.

**Writing – review & editing:** Farah Nassar, Mehdi Bourouba, Nasr Rihab.

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
