## [Decision Letter · Decision Letter 0]

7 Oct 2019

PONE-D-19-22404

miRNA Expression in Advanced Algerian Breast Cancer Tissues

PLOS ONE

Dear Dr. Nasr,

Thank you for submitting your manuscript to PLOS ONE. After careful consideration, we feel that it has merit but does not fully meet PLOS ONE’s publication criteria as it currently stands. Therefore, we invite you to submit a revised version of the manuscript that addresses the points raised during the review process.

We would appreciate receiving your revised manuscript by Nov 21 2019 11:59PM. To enhance the reproducibility of your results, we recommend that if applicable you deposit your laboratory protocols in protocols.io, where a protocol can be assigned its own identifier (DOI) such that it can be cited independently in the future. For instructions see: http://journals.plos.org/plosone/s/submission-guidelines#loc-laboratory-protocols

We look forward to receiving your revised manuscript.

Kind regards,

George Calin

Academic Editor

PLOS ONE

Journal Requirements:

1. We suggest you thoroughly copyedit your manuscript for language usage, spelling, and grammar. If you do not know anyone who can help you do this, you may wish to consider employing a professional scientific editing service.  

2. Thank you for including your funding statement; "The funders had no role in study design, data collection and analysis, decision to publish, or preparation of the manuscript."

Please provide an amended Funding Statement that declares *all* the funding or sources of support received during this specific study (whether external or internal to your organization) as detailed online in our guide for authors at http://journals.plos.org/plosone/s/submit-now.  

Please state what role the funders took in the study.  If any authors received a salary from any of your funders, please state which authors and which funder. If the funders had no role, please state: "The funders had no role in study design, data collection and analysis, decision to publish, or preparation of the manuscript."

Reviewers' comments:

Reviewer's Responses to Questions

**Comments to the Author**

1. Is the manuscript technically sound, and do the data support the conclusions?

Reviewer #1: Yes

Reviewer #2: No

2. Has the statistical analysis been performed appropriately and rigorously? 

Reviewer #1: Yes

Reviewer #2: N/A

3. Have the authors made all data underlying the findings in their manuscript fully available?

Reviewer #1: Yes

Reviewer #2: No

4. Is the manuscript presented in an intelligible fashion and written in standard English?

Reviewer #1: Yes

Reviewer #2: Yes

5. Review Comments to the Author

Reviewer #1: In the manuscript titled “miRNA Expression in Advanced Algerian Breast Cancer Tissues” Tfaily, Nassar et al. analyze the expression of several miRNAs from breast cancer and adjacent normal tissue from Algerian women in order to find an ethnic specific miRNA signature.

Remarks:

1. Details about the miRNA biogenesis are redundant, can be removed.

2. Introduction, use the correct nomenclature for miRNA, so instead of miRNAs 142 and 145, use miR-143 and miR-145.

3. Provide a table in which you present for each of the miRNAs selected for your study the deregulation on other populations + references and in the last column present your findings. How much is specific for Algerians and how much is overlapping with other ethnicities?

4. Not very clear why you selected these 7 miRNAs? Make it more clear by bringing arguments from the literature, see also remark number 3.

5. Provide a second table in which you present the baseline demographics. Would be interesting to know for which of the 22 patients you had also the adjacent normal tissue.

6. The first three lines from point 3.2. belong to the methods section.

7. Figure 1 and figure 2 add the P-value in the figure and also in the manuscript. The star seems to be a dot in the tumor group.

8. Figure 3 is incompletely labeled. What does the yellow and gray signify?

9. Provide data regarding the levels of U6 (cycles) in normal and tumor samples.

10. Very strange you obtained the same P-value for miR-21 and miR-125b for HER2 analysis (0.021)?

11. Point 3.3. not clear which is the purpose of analyzing the correlation between miRNAs. Did you check the correlations only in tumor tissue or both in normal and tumor? One possible way to use this date is to build miRNA networks, where to miRNAs are connected if they correlate (check the fallowing manuscripts in order to understand this concept: PMID: 20439436; PMID: 29949872 and PMID: 28820886). Of corse it is not correct to build miRNA networks separately for normal and tumor because there are not enough samples in the normal group. Hence, build a miRNA network for all samples combined or only for breast cancer.

12. Add a sub-chapter 3.4. in which you analyze systematically the data you discovered and the data discovered by others regarding these miRNAs in breast cancer.

13. Pathway analysis for the targets of the up-regulated miRNAs is also necessary in order to gain mechanistic insights in the function of this miRNAs.

Overall I consider this paper incomplete and multiple additional analysis are necessary.

Reviewer #2: The present article presents the expression of several miRNAs in tissue samples from Algerian breast cancer patients that are proposed as possible biomarkers in the ethnic context. Moreover is highlighted the fact that Algerian women are diagnosed at a much earlier age compared to other regions, but no explanation from a miRNA point of view is offered in this context. Also, the authors state that there is a need for new screening methods due to inaccessibility to the current ones; however, a screening based on miRNA requires advanced infrastructure and training. Moreover, because miRNAs are involved in multiple processes, it is hard to associate their aberrant expression with a specific pathology (even differentiate between cancer and other diseases/conditions). Another significant drawback is represented by the small number of samples in the context of a very heterogeneous disease

- Authors affirm that the miRNA signature can be used as a biomarker for the Algerian population; however, these miRNAs were repeatedly found in other studies on breast cancer. Before affirming the ethnic specificity, authors should comprehensively analyze the results obtained on other types of populations

- In the introduction, authors should update the information according to new Globocan data (not the one from 2012)

- “The young median age at diagnosis, high incidence and mortality rates and the inaccessibility of current screening tools of breast cancer highlight the importance of novel screening techniques and early detection in decreasing the morbidity and mortality of the disease” – parts of this phrase are somehow contradictory; as the authors suggest at the moment there is a problem with the accessibility to consecrated screening tools, but there is much easier to detect some miRNAs as biomarkers for diagnosis? Usually, the detection of miRNAs requires a somehow complex infrastructure, and these sequences are also quite unspecific due to the extensive involvement in different processes.

- Is not entirely clear how the authors selected the following miRNAs: miR-21, miR-425-5p, miR-183, miR-182, miR-200c, miR-125b and miR-100; moreover, if the authors are aiming to select a profile of miRNAs specific for the Algerian population, they should not choose the most common one found in literature and may be investigated first the whole profile of miRNAs in several samples

- Authors state that they are looking for reliable and accessible screening methods, but the miRNAs are analyzed from tissues biopsy where the anatomopathological exam is the most trustful

- Breast cancer has a significant number of subtypes according to the expression of the hormone receptors; the author included a very restrictive number of patients and is impossible to analyze the expression of these miRNAs in concordance with the breast cancer subtype – miRNAs vary quite significantly between the different subtypes

6. PLOS authors have the option to publish the peer review history of their article (what does this mean?). If published, this will include your full peer review and any attached files.

Reviewer #1: No

Reviewer #2: No

---

## [Author Response · Author response to Decision Letter 0]

25 Nov 2019

Kindly check the attached file "Response to reviewers" as we couldn't upload all document here;

Date: 20 Nov 2019

Subject: Rebuttal Letter

Manuscript ID: PONE-D-19-22404

Manuscript title: miRNA Expression in Advanced Algerian Breast Cancer Tissues

Dear Dr. Calin, 

We would like to thank the editors and the reviewers for the valuable suggestions and comments and for the time and efforts taken in reviewing our manuscript.

Kindly find below a point-by-point response to all the comments raised by the reviewers. We are also attaching a revised version of the manuscript that highlights changes made to the original version and an unmarked version of our revised paper without tracked changes.

We hope that you will find our justifications sufficient to consider our manuscript for publication in PloS One.

Best regards,

Rihab Nasr

Dr. Rihab Nasr

Associate Professor

Department of Anatomy, Cell Biology and Physiological Sciences

Director of Basic Research Core Facilities

Director of Cancer Prevention and Control Program

Founder of AMALOUNA 

Faculty of Medicine

American University of Beirut

Beirut - Lebanon

Phone: 01 350000Ext: 4812

Subject: Rebuttal letter 

Manuscript ID: PONE-D-19-22404

Manuscript Title: miRNA Expression in Advanced Algerian Breast Cancer Tissues

Reviewer # 1 raised points that we would like to clarify:

In the manuscript titled “miRNA Expression in Advanced Algerian Breast Cancer Tissues” Tfaily, Nassar et al. analyze the expression of several miRNAs from breast cancer and adjacent normal tissue from Algerian women in order to find an ethnic specific miRNA signature.

1. Details about the miRNA biogenesis are redundant, can be removed.

Thank you for your remark. As per your recommendation, the redundant information are now omitted in the revised manuscript.

2. Introduction, use the correct nomenclature for miRNA, so instead of miRNAs 142 and 145, use miR-143 and miR-145.

Thank you for your note. The nomenclature has been fixed now in the revised manuscript. 

3. Provide a table in which you present for each of the miRNAs selected for your study the deregulation on other populations + references and in the last column present your findings. How much is specific for Algerians and how much is overlapping with other ethnicities?

As per the reviewer recommendation, we now added the below table that shows different studies on the selected miRNA along with their mode of deregulation in specific populations/countries and the following text to the revised manuscript in section

“3.4 miRNA deregulation in our study compared to the literature 

miRNA deregulation in the Algerian population was compared to that found from several other studies in the literature (Table 1). Our results revealed upregulation of miR-21, miR-183, miR-182, miR-425-5p and miR-200c that is in concordance with the results across the literature. Nevertheless, our study revealed non-significant upregulation of miR-125b and miR-100 while results in the literature reveal downregulation of these two miRNAs.” 

. 

miRNA Deregulation in Our study Deregulation in Literature Population/Country

if available Reference 

miR-183 Up Up Lebanese, American 1, 3, 5

miR-182 Up Up Lebanese, Chinese 1, 6, 11

miR-125b Up (non-significant) Down Lebanese, Spanish, Taiwan 1, 4, 7

miR-100 Up (non-significant) Down Spanish, Austria 7, 8, 10

miR-200c Up Up Lebanese, American, Taiwan 1,3,4, 10

miR-425-5p Up Up Lebanese, American 1, 3

miR-21 Up Up Italy, USA, Taiwan 2, 3, 4, 9

Table 1. Mode of deregulation of the studied miRNA in different studies in comparison to our findings. (Up means upregulated; Down means downregulation).

Table references:

1. Nassar FJ, Talhouk R, Zgheib NK, Tfayli A, El Sabban M, El Saghir NS, et al. microRNA Expression in Ethnic Specific Early Stage Breast Cancer: an Integration and Comparative Analysis. Sci Rep. 2017;7(1):16829.

2. Iorio MV, Ferracin M, Liu CG, Veronese A, Spizzo R, Sabbioni S, et al. MicroRNA gene expression deregulation in human breast cancer. Cancer Res. 2005;65(16):7065-70.

3. Chen L, Li Y, Fu Y, Peng J, Mo MH, Stamatakos M, et al. Role of deregulated microRNAs in breast cancer progression using FFPE tissue. PLoS One. 2013;8(1):e54213.

4. Tsai HP, Huang SF, Li CF, Chien HT, Chen SC. Differential microRNA expression in breast cancer with different onset age. PLoS One. 2018;13(1):e0191195.

5. Nassar FJ, El Sabban M, Zgheib NK, Tfayli A, Boulos F, Jabbour M, et al. miRNA as potential biomarkers of breast cancer in the Lebanese population and in young women: a pilot study. PLoS One. 2014;9(9):e107566.

6. Wang PY, Gong HT, Li BF, Lv CL, Wang HT, Zhou HH, et al. Higher expression of circulating miR-182 as a novel biomarker for breast cancer. Oncol Lett. 2013;6(6):1681-6.

7. Matamala N, Vargas MT, González-Cámpora R, Miñambres R, Arias JI, Menéndez P, et al. Tumor microRNA expression profiling identifies circulating microRNAs for early breast cancer detection. Clin Chem. 2015;61(8):1098-106.

8. Gebeshuber CA, Martinez J. miR-100 suppresses IGF2 and inhibits breast tumorigenesis by interfering with proliferation and survival signaling. Oncogene. 2013;32(27):3306-10.

9. Song, B., Wang, C., Liu, J. et al. MicroRNA-21 regulates breast cancer invasion partly by targeting tissue inhibitor of metalloproteinase 3 expression. J Exp Clin Cancer Res 29, 29 (2010) doi:10.1186/1756-9966-29-29

10. Gong, Y., He, T., Yang, L. et al. The role of miR-100 in regulating apoptosis of breast cancer cells. Sci Rep 5, 11650 (2015) doi:10.1038/srep11650

11. Chi-Hsiang Chiang, Ming-Feng Hou, Wen-Chun Hung et al. Up-regulation of miR-182 by β-catenin in breast cancer increases tumorigenicity and invasiveness by targeting the matrix metalloproteinase inhibitor RECK, Biochimica et Biophysica Acta (BBA) (2012)

https://doi.org/10.1016/j.bbagen.2013.01.009.

4. Not very clear why you selected these 7 miRNAs? Make it more clear by bringing arguments from the literature, see also remark number 3.

The reviewer has raised an important point regarding the choice of the tested miRNAs. These 7 miRNAs were selected based on a miRNA microarray recently performed on formalin fixed paraffin embedded tissues from Lebanese breast cancer patients by Nassar et al. (2017). This is the only array available for breast cancer tissues in the MENA for the Arab population. Those miRNAs were also validated using real time PCR on the breast cancer tissues of Lebanese patients. miR-183, miR-182, miR-200c, miR-425-5p, miR-21 were significantly upregulated and miR-125b and 100 were significantly downregulated in tumor versus normal adjacent tissues. We have now added this justification in the last paragraph of the introduction in the revised manuscript.

5. Provide a second table in which you present the baseline demographics. Would be interesting to know for which of the 22 patients you had also the adjacent normal tissue.

As per your recommendation, kindly find below the table that presents the baseline demographics of the samples.

Sample Number Age HR Status HER2 Presence of Normal Tissue

Tumor 1 69 ER+/PR+ Positive No

Tumor 2 33 ER+/PR+ Negative No

Tumor 3 48 ER+/PR+ Positive Yes

Tumor 4 60 ER+/PR+ Positive No

Tumor 5 78 ER+/PR+ Positive No

Tumor 6 38 ER-/PR- Positive No

Tumor 7 62 ER+/PR+ Positive No

Tumor 8 43 ER+/PR+ Negative No

Tumor 9 41 ER+/PR+ Positive No

Tumor 10 44 ER-/PR- Negative Yes

Tumor 11 77 ER-/PR- Positive No

Tumor 12 NA NA NA Yes

Tumor 13 50 ER+/PR+ Negative Yes

Tumor 14 57 ER-/PR- Positive No

Tumor 15 61 ER-/PR- Positive No

Tumor 16 37 ER+/PR+ Negative No

Tumor 17 49 ER+/PR+ Negative No

Tumor 18 NA NA NA Yes

Tumor 19 35 ER+/PR+ Negative No

Tumor 20 42 ER+/PR+ Negative No

Tumor 21 60 ER+/PR+ Negative Yes

Tumor 22 70 ER+/PR+ Positive No

Table. Clinical Characteristics of breast cancer tumor tissues from Algerian patients

6. The first three lines from point 3.2. belong to the methods section.

Thank you for this note. The manuscript was changed accordingly and the first three lines from point 3.2 are deleted. 

7. Figure 1 and figure 2 add the P-value in the figure and also in the manuscript. The star seems to be a dot in the tumor group.

The p-values have been added to the figures and the manuscript. The star sign has been enlarged to be more visible.

Updated Figure 1

Updated Figure 2

8. Figure 3 is incompletely labeled. What does the yellow and gray signify?

Thank you for the observation. The grey label highlights the tumor samples and the yellow highlights the normal adjacent tissue. To comply with the reviewer’s comment a clarifying mention has been added to figure 3 legend on page 12 in revised manuscript.

9. Provide data regarding the levels of U6 (cycles) in normal and tumor samples.

Kindly find the figure below that shows a bar graph of the cycles of RNU6B in tumor and normal adjacent tissue samples. The average of the cycles is 30.92 and SEM is 0.3258. 

10. Very strange you obtained the same P-value for miR-21 and miR-125b for HER2 analysis (0.021)?

Mann-whitney test was performed to compare both HER2 negative and positive groups. The p-value for miR-21 and miR-125b is not the same (0.0031 for miR-21 and 0.031 for miR-125b). Below is screenshot of the statistical p-value that we got using GraphPad Prism.

We apologize for this mistake and we thank the reviewer for pointing it out. This is now corrected in the revised manuscript, section 3.2.

 miR-125b miR-21

11. Point 3.3. not clear which is the purpose of analyzing the correlation between miRNAs. Did you check the correlations only in tumor tissue or both in normal and tumor? One possible way to use this date is to build miRNA networks, where to miRNAs are connected if they correlate (check the fallowing manuscripts in order to understand this concept: PMID: 20439436; PMID: 29949872 and PMID: 28820886). Of corse it is not correct to build miRNA networks separately for normal and tumor because there are not enough samples in the normal group. Hence, build a miRNA network for all samples combined or only for breast cancer.

The correlations were formed based on the tumor tissue analysis, and no normal tissue correlations were done. We thank the reviewer for the detailed suggestion. Because this is a pilot study and our sample size is small, we will consider using these suggested correlations in our future studies with larger sample size. However, in reply to the below comment, we have already used the Enricher bioinformatics resources suggested by the reviewer to get the GO Biological pathways of validated targets for the studied miRNAs. We have also performed Network Analysis between the miRNAs using Pathway Studio and demonstrated that the studied miRNA (miR-125b, 183, 182, 21, 125b, 200c and 425-5p) are involved in several vital biological processes of breast cancer. Please check our reply to comment 13.

12. Add a sub-chapter 3.4. in which you analyze systematically the data you discovered and the data discovered by others regarding these miRNAs in breast cancer.

Thank you for this comment. As explained in our reply to comment 3, this is now added as a table 1 and the new section: 3.4 miRNA deregulation in our study compared to the literature 

miRNA deregulation in the Algerian population was compared to that found from several other studies in the literature (Table 1). Our results revealed upregulation of miR-21, miR-183, miR-182, miR-425-5p and miR-200c that is in concordance with the results across the literature. Nevertheless, our study revealed insignificant upregulation of miR-125b and miR-100 while results in the literature reveal downregulation of these two miRNAs.” 

13. Pathway analysis for the targets of the up-regulated miRNAs is also necessary in order to gain mechanistic insights in the function of this miRNAs.

We thank the reviewer for this important suggestion. We now performed Network Analysis between the miRNAs using Pathway Studio which enables the analysis and visualization of altered pathways required to construct and recognize altered cellular processes and involved molecular functional pathways in breast cancer. This led us to demonstrate that the studied miRNA (miR-125b, 183, 182, 21, 125b, 200c and 425-5p) are involved in several vital biological processes of breast cancer which are cell differentiation, cell motility, cell death, oncogenesis, cell invasion and migration, DNA damage and chemosensitivity. This is now added in the first paragraph of the discussion and presented as Figure 5. Moreover, GO Biological Processes of validated targets for miR-182,183, 100, 21, 200c and 125b were also determined using Enrichr software (https://amp.pharm.mssm.edu/Enrichr/enrich). Validated targets were identified using Pubmed literature search and their GO Biological Processes were determined using this software and presented as supplementary figure 1. 

Figure 5. Pathway Studio Network analysis of the studied microRNA in breast cancer. A biological network was created using the Pathway Studio 9.0 program to visualize the role of different microRNA discussed in this manuscript in chosen breast cancer-related pathways.

Reviewer #2 raised the below points that we would like to clarify:

1. The present article presents the expression of several miRNAs in tissue samples from Algerian breast cancer patients that are proposed as possible biomarkers in the ethnic context. Moreover is highlighted the fact that Algerian women are diagnosed at a much earlier age compared to other regions, but no explanation from a miRNA point of view is offered in this context. 

We thank the reviewer for these comments. While it is true that Algerian women are diagnosed at a much earlier age, the low sample size in our study, hindered forming a correlation between patient age and miRNA deregulation. We hope this to be a pilot study for further studies to follow studying miRNA deregulation according to age of breast cancer patients. 

2. Also, the authors state that there is a need for new screening methods due to inaccessibility to the current ones; however, a screening based on miRNA requires advanced infrastructure and training. 

We agree with the reviewer that screening based on miRNA requires advanced infrastructure and training, but this is the same with all novel technologies. Even mammography has a lot challenges regarding its accessibility and analysis. A mobile mammography service is currently being employed by Roche and Algerian government for earlier screening to populations in rural areas, but more optimal solutions can be researched and applied especially that mammography method excludes young patients and not all people have access to it in such a vast country, Algeria. 

Current research is focusing on liquid biopsies for disease detection such as biomarkers in blood and serum for early detection of breast cancer. With technological advancements and genetics and cancer research abundance, these modalities are expected to become more accessible in the coming decades (Sherefatian et al., 2018; Ivanov et al., 2018; Blenkiron et al.; 2007). Interestingly, miRNAs are present in several biological fluids including blood, plasma, serum. miRNAs are abundant, nuclease-resistant and consistently quantifiable in sera of individuals of the same species. A key advantage of miRNA is the easiness of their detection using microarray, deep sequencing or reverse transcription quantitative real-time PCR (RT-qPCR). Hence, being stable, non-invasive, specific and measurable makes miRNA ideal biomarkers for cancer diagnosis, prognosis and therapy prediction. This pilot study aims to lay ground to such research in Algeria and North Africa, as transporting blood samples is expected to be much easier than transportation of radiological machines, with the advancement of PCR technology and other biosensors. 

This is now addressed in the last paragraph of the discussion.

3. Moreover, because miRNAs are involved in multiple processes, it is hard to associate their aberrant expression with a specific pathology (even differentiate between cancer and other diseases/conditions). Another significant drawback is represented by the small number of samples in the context of a very heterogeneous disease

While this has not been able to be fully explored in our study due to the low sample size, miRNA association with specific pathologies is widely available in the literature. We fully agree with the reviewer that although it may be true that a single miRNA profiling may not be able to distinguish between different diseases, however, the future aim is using panels of miRNAs that will be able to screen, diagnose and predict prognosis and response to treatment in breast cancer. For example, an early study investigating miRNA deregulation in the blood of breast cancer patients was conducted by Chang et al. in 2015. Using HiSeq 2500, miRNAs like miR-144-3p, miR-451a and miR-144-5p were found to be upregulated in peripheral blood mononuclear cells (PBMC) with fold changes ranging between 2.61 and 3.05. miR-708-5p was found to be downregulated in the PBMC by a fold change of 0.46 (Chang et al. 2015). It is worth noting that miR-195-5p and miR-495 in patient PBMCs had a specifity and sensitivity of 100%, respectively, enabling them to be valuable diagnostic tools (Mishra et al., 2015). Moreover, Fang et al. reported combinations of miRNAs in patient plasma that were able to detect breast cancer with an AUC of 0.931 when compared with the normal group (Fang et al., 2019). 

4. Authors affirm that the miRNA signature can be used as a biomarker for the Algerian population; however, these miRNAs were repeatedly found in other studies on breast cancer. Before affirming the ethnic specificity, authors should comprehensively analyze the results obtained on other types of populations

Two meta-analysis done revealed variated miRNA deregulation by ethnicity (Chen et al., 2014; Wang et al., 2012). Most of the miRNA studies were done on Eurasian and American populations. A study on the Lebanese population was done by Farah et al. in 2016, by which a microarray of miRNA revealed several miRNAs deregulated in breast cancer with some variation when compared to American population. Given the lack of microarrays on the North African/Algerian population, we used the microarray done on Lebanese populations to choose the corresponding miRNAs to study in the Algerian population. 

Although in the study we conclude that these miRNA deregulations are expected to be generalized to the Algerian population, we do not imply specificity to the Algerian ethnicity. To the contrary, we found that most of the deregulations were in harmony with those found in other populations around the globe, with the exception of miR-100 and miR-125b whose results were inconclusive. 

5. In the introduction, authors should update the information according to new Globocan data (not the one from 2012)

We thank the reviewer for this note. Reference 2 and information have been updated accordingly. 

6. “The young median age at diagnosis, high incidence and mortality rates and the inaccessibility of current screening tools of breast cancer highlight the importance of novel screening techniques and early detection in decreasing the morbidity and mortality of the disease” – parts of this phrase are somehow contradictory; as the authors suggest at the moment there is a problem with the accessibility to consecrated screening tools, but there is much easier to detect some miRNAs as biomarkers for diagnosis? Usually, the detection of miRNAs requires a somehow complex infrastructure, and these sequences are also quite unspecific due to the extensive involvement in different processes.

We thank the reviewer for his/her comment and we agree that that screening based on miRNA requires advanced infrastructure and training. This is now highlighted in the last paragraph of the discussion in the revised manuscript:

“The aim of this pilot study was to lay ground to microRNA research in the region. As such, further global miRNA microarray profiling can aid in finding ethnic specific miRNA biomarkers in the Algerian breast cancer population. Moreover, further research on circulating microRNA is needed which along with the availability of advanced infrastructure, technology and training might serve identifying potential biomarkers for early detection of breast cancer”.

7. Is not entirely clear how the authors selected the following miRNAs: miR-21, miR-425-5p, miR-183, miR-182, miR-200c, miR-125b and miR-100; moreover, if the authors are aiming to select a profile of miRNAs specific for the Algerian population, they should not choose the most common one found in literature and may be investigated first the whole profile of miRNAs in several samples

The reviewer has raised an important point regarding the choice of miRNA. These 7 miRNAs were selected based on a miRNA microarray performed on formalin fixed paraffin embedded tissues from Lebanese breast cancer patients by Nassar et al. (2017). This is the only array available for breast cancer tissues in the MENA for the Arab population. Those miRNAs were also validated using real time PCR on the breast cancer tissues of Lebanese patients. miR-183, miR-182, miR-200c, miR-425-5p, miR-21 were significantly upregulated and miR-125b and miR-100 were significantly downregulated in tumor versus normal adjacent tissues. We have now added this justification in the last paragraph of the introduction in the revised manuscript.

8. Authors state that they are looking for reliable and accessible screening methods, but the miRNAs are analyzed from tissues biopsy where the anatomopathological exam is the most trustful

Thank you for this comment. We fully agree with the reviewer that the anatomopathological exam is indeed the most reliable in diagnosing breast cancer. Our current study aims to explore the deregulation of miRNAs in the tissues to provide more insight into the process of carcinogenesis. However, our future studies aim at exploring circulating miRNAs deregulation in the blood samples of breast cancer patients, considered as easily accessible and less invasive samples. We have now added “circulating” to microRNA when discussing them as biomarkers in the introduction.

“Novel biomarkers including circulating miRNA levels are under extensive study for their potential role in being effective screening tools of this disease.”

9. Breast cancer has a significant number of subtypes according to the expression of the hormone receptors; the author included a very restrictive number of patients and is impossible to analyze the expression of these miRNAs in concordance with the breast cancer subtype – miRNAs vary quite significantly between the different subtypes. 

Given this being a pilot study with small sample size, doing subgroup analysis is indeed difficult to analyze and generalize. Nevertheless, we aim to help future large scale studies to base their miRNA decision and a priori hypotheses based on our results to yield more accurate and precise results. 

We finally thank the editor and the reviewers for the valuable suggestions and comments and we hope that the above clarifications will meet with approval

Rihab Nasr

References: 

Blenkiron C, Miska EA (2007) miRNAs in cancer: approaches, diagnostics and therapy and

therapy. Hum Mol Genet 16: 106-113.

Chang, C.W., et al., microRNA Expression in Prospectively Collected Blood as a Potential

Biomarker of Breast Cancer Risk in the BCFR. Anticancer Res, 2015. 35(7): p. 3969-77.

Chen QH, Wang QB, Zhang B. Ethnicity modifies the association between functional microRNA

polymorphisms and breast cancer risk: a Huge meta-analysis. Tumour Biol. 2014;35(1):529-43.

Fang, R., et al., Plasma MicroRNA Pair Panels as Novel Biomarkers for Detection of Early Stage Breast Cancer. Frontiers in Physiology, 2019. 9(1879).

Ivanov YD, Pleshakova TO, Malsagova KA, Kozlov AF, Kaysheva AL, et al. (2018) Detection

of marker miRNAs in plasma using SOI-NW biosensor. Sens Actuators B Chem 261: 566-571. 

Mishra, S., et al., Circulating miRNAs revealed as surrogate molecular signatures for the early detection of breast cancer. Cancer Letters, 2015. 369(1): p. 67-75.

Nassar FJ, El Sabban M, Zgheib NK, Tfayli A, Boulos F, Jabbour M, et al. miRNA as potential

biomarkers of breast cancer in the Lebanese population and in young women: a pilot study. PLoS One. 2014;9(9):e107566.

Nassar FJ, Talhouk R, Zgheib NK, Tfayli A, El Sabban M, El Saghir NS, et al. microRNA Expression in Ethnic Specific Early Stage Breast Cancer: an Integration and Comparative

Analysis. Sci Rep. 2017;7(1):16829.

Roche, Mobile breast cancer service

https://www.roche.com/sustainability/access-to-healthcare/ath_bc_algeria.htm

Sherafatian et al., Tree-based machine learning algorithms identified minimal set of miRNA

biomarkers for breast cancer diagnosis and molecular subtyping. Gene, 2018. 677: 111-118. 

Wang AX, Xu B, Tong N, Chen SQ, Yang Y, Zhang XW, et al. Meta-analysis confirms that a

common G/C variant in the pre-miR-146a gene contributes to cancer susceptibility and that ethnicity, gender and smoking status are risk factors. Genet Mol Res. 2012;11(3):3051-62.

---

## [Decision Letter · Decision Letter 1]

3 Jan 2020

miRNA Expression in Advanced Algerian Breast Cancer Tissues

PONE-D-19-22404R1

Dear Dr. Nasr,

We are pleased to inform you that your manuscript has been judged scientifically suitable for publication and will be formally accepted for publication once it complies with all outstanding technical requirements.

With kind regards,

George Calin

Academic Editor

PLOS ONE

Additional Editor Comments (optional):

Reviewers' comments:

Reviewer's Responses to Questions

**Comments to the Author**

1. If the authors have adequately addressed your comments raised in a previous round of review and you feel that this manuscript is now acceptable for publication, you may indicate that here to bypass the “Comments to the Author” section, enter your conflict of interest statement in the “Confidential to Editor” section, and submit your "Accept" recommendation.

Reviewer #1: All comments have been addressed

2. Is the manuscript technically sound, and do the data support the conclusions?

Reviewer #1: Yes

3. Has the statistical analysis been performed appropriately and rigorously? 

Reviewer #1: Yes

4. Have the authors made all data underlying the findings in their manuscript fully available?

Reviewer #1: Yes

5. Is the manuscript presented in an intelligible fashion and written in standard English?

Reviewer #1: Yes

6. Review Comments to the Author

Reviewer #1: The manuscript has improved significantly after the first submission and I consider it reaches the high standards of the journal. No additional modifications are necessary.

7. PLOS authors have the option to publish the peer review history of their article (what does this mean?). If published, this will include your full peer review and any attached files.

Reviewer #1: No

---

## [Editor Report · Acceptance letter]

16 Jan 2020

PONE-D-19-22404R1 

miRNA Expression in Advanced Algerian Breast Cancer Tissues 

Dear Dr. Rihab:

I am pleased to inform you that your manuscript has been deemed suitable for publication in PLOS ONE. Congratulations! Your manuscript is now with our production department. 

With kind regards,

on behalf of

Dr. George Calin 

Academic Editor

PLOS ONE